# Lack of the Histone Deacetylase SIRT1 Leads to Protection against Endoplasmic Reticulum Stress through the Upregulation of Heat Shock Proteins

**DOI:** 10.3390/ijms25052856

**Published:** 2024-03-01

**Authors:** Jessica Latorre, Nuria de Vera, Tomàs Santalucía, Rafel Balada, Anna Marazuela-Duque, Alejandro Vaquero, Anna M. Planas, Valérie Petegnief

**Affiliations:** 1Department of Neuroscience and Experimental Therapeutics, Institute for Biomedical Research of Barcelona (IIBB), Spanish Research Council (CSIC), 08036 Barcelona, Spain; jessica.latorre@uvic.cat (J.L.); ndevera64@gmail.com (N.d.V.); rafael.baladacaballe@unipd.it (R.B.); anna.planas@iibb.csic.es (A.M.P.); 2Department of Fundamental and Clinical Nursing, School of Nursing, University of Barcelona, 08907 L’Hospitalet de Llobregat, Spain; tomas.santalucia@ub.edu; 3Chromatin Biology Laboratory, Josep Carreras Leukaemia Research Institute, 08916 Badalona, Spain; amarazuela@carrerasresearch.org (A.M.-D.); avaquero@carrerasresearch.org (A.V.); 4Institut d’Investigacions Biomèdiques August Pi Sunyer (IDIBAPS), 08036 Barcelona, Spain

**Keywords:** sirtuin, Hsp70, STAT3, ER stress

## Abstract

Histone deacetylase SIRT1 represses gene expression through the deacetylation of histones and transcription factors and is involved in the protective cell response to stress and aging. However, upon endoplasmic reticulum (ER) stress, SIRT1 impairs the IRE1α branch of the unfolded protein response (UPR) through the inhibition of the transcriptional activity of XBP-1 and SIRT1 deficiency is beneficial under these conditions. We hypothesized that SIRT1 deficiency may unlock the blockade of transcription factors unrelated to the UPR promoting the synthesis of chaperones and improving the stability of immature proteins or triggering the clearance of unfolded proteins. SIRT1+/+ and SIRT1−/− fibroblasts were exposed to the ER stress inducer tunicamycin and cell survival and expression of heat shock proteins were analyzed 24 h after the treatment. We observed that SIRT1 loss significantly reduced cell sensitivity to ER stress and showed that SIRT1−/− but not SIRT1+/+ cells constitutively expressed high levels of phospho-STAT3 and heat shock proteins. Hsp70 silencing in SIRT1−/− cells abolished the resistance to ER stress. Furthermore, accumulation of ubiquitinated proteins was lower in SIRT1−/− than in SIRT1+/+ cells. Our data showed that SIRT1 deficiency enabled chaperones upregulation and boosted the proteasome activity, two processes that are beneficial for coping with ER stress.

## 1. Introduction

Sirtuins are Class III NAD+-dependent histone deacetylases (HDAC) that control gene expression and are involved in the response to a variety of stress stimuli. Sirtuins regulate a myriad of proteins, thereby restoring cell homeostasis under cell disturbances such as nutrient depletion, dysfunctional glucose and lipid metabolism and oxidative stress [1]. Silent mating type information regulation 2 homolog (SIRT1) is the most studied member of the sirtuins family and plays a key role in cellular adaptive processes and aging [1]. Substrates for SIRT1 include not only histones but also transcription factors and cytoplasmic proteins. SIRT1 levels, its subcellular localization and enzymatic activity are critically regulated to provide the cell with the appropriate response against each specific stress stimulus [2]. Indeed, increased activity of SIRT1 improves cell outcome in ischemia-reperfusion in liver and neurodegenerative diseases [3,4] through the deacetylation and activation of the transcription factor peroxisome proliferator-activated receptor-gamma coactivator (PGC1) and inhibition of nuclear factor kappa B (NF-κB) amongst others. In contrast, uncontrolled SIRT1 activity strengthens cancer survival [2], promotes the pro-inflammatory phenotype of astrocytes in autoimmune disease through the inhibition of Nrf2 targets [5] and it is harmful in some hypoxic stress conditions because it exacerbates NAD+ consumption [6,7]. Therefore, it is essential to analyze the beneficial and detrimental effects of SIRT1 activation in cell stress models. Wang and collaborators [8] described that, in SIRT1−/− mouse embryonic fibroblasts (MEFs), X-box binding protein 1 (XBP-1) was acetylated and displayed an enhanced transcriptional activity, thereby providing a robust protection against ER stress. These authors concluded that SIRT1 is a negative regulator of the unfolded protein response (UPR). Since ER stress contributes to many metabolic and neurodegenerative diseases involving the accumulation of unfolded proteins, it is physiologically relevant to understand how cells with impaired or deficient SIRT1 activity, a situation described in aged cells and after chronic stress [9], adapt to cope with stressful conditions and to identify the mediators of the protection. This will also help to design strategies to finely tune the use of SIRT1 regulators.

ER stress promotes the accumulation of unfolded proteins and at the same time, the synthesis of immunoglobulin heavy chain binding protein (BiP or Grp78) to bind to and avoid the degradation of these immature proteins as part of the UPR. Inhibitors of class I and class IIa HDAC upregulate protective chaperones such as Bip and heat shock protein 70 (Hsp70) [10,11,12,13] through the deacetylation of specificity protein 1 (Sp1) [10], although there are no data regarding chaperones induction after treatment with class III HDAC (sirtuins) inhibitors. Accumulation of heat shock proteins (HSP) occurs following many cell stress situations including ischemia-hypoxia [14], heat shock [15], chemical poisoning of the mitochondrial respiratory chain [16] or inhibition of the proteasome [17] mainly through the increased activity of heat shock factor 1 (HSF-1) [17]. Signal transducer and activator of transcription 1 and 3 (STAT1 and STAT3) are transcription factors known to regulate Hsp70 expression in human cells stimulated with cytokines [18] or overexpression of Hsp105β [19], respectively, but there are no data in murine cells. Interestingly, STAT3 is deacetylated and inhibited by SIRT1 [20] and HSF-1 activity is also regulated by SIRT1 [21].

SIRT1, through the interaction with tuberous sclerosis complex (TSC-2), blocks mammalian target of rapamycin (mTOR) activity [22]. SIRT1−/− cells have an elevated rate of basal de novo protein synthesis due to an increased activity of the mTORC1 kinase that phosphorylates eukaryotic translation initiation factor 4-binding protein (4E-BP1) and ribosomal protein S6 kinase (S6K1) [22]. However, it is unclear whether the increased protein synthesis rate requires an extra-production of chaperones. mTORC1 is a master regulator of cell viability. It stimulates translation, cell proliferation and inhibits apoptosis and autophagy [23]. Interestingly, STAT3 phosphorylation by mTORC1 promotes cell survival [24]. Therefore, mTORC1 overactivation due to SIRT1 loss of function could also contribute to SIRT1−/− cell resistance to ER stress.

In the present paper, we explored several related signaling pathways that could explain how SIRT1 deficiency reduces cell death induced by ER stress, such as mTOR activation, the expression of chaperones or the degradation of unfolded proteins and the transcriptional activity of HSE and STAT3 response elements in SIRT1+/+ and SIRT1−/− mouse fibroblasts. We showed that Hsp70 plays a critical role in the resilience of fibroblasts lacking SIRT1 against an ER stress. Some of the results were in agreement with those observed by other authors. Yet, our study went a step further, and provided evidence showing that Hsp70 expression in SIRT1−/− cells in basal conditions is independent of transcription factors acting on heat shock element (HSE). Here we revealed an integrative role for Hsp70, that assists the folding of newly synthesized proteins in physiological conditions and avoids the accumulation of unfolded proteins produced by ER stress in SIRT1 deficient fibroblasts.

## 2. Results

### 2.1. SIRT1-Deficient Cells Were Less Vulnerable to ER Stress Than wt Cells but Developed Similar UPR, and Resistance Was Independent of mTOR Hyperactivity

We first validated a model of tunicamycin-induced ER stress and cell damage. To monitor cell survival, we counted the number of nuclei stained with DAPI 24 h after the treatment with tunicamycin. At that time, dead cells had detached. Tunicamycin induced a loss of 71% and 42% of the cells in wt and SIRT1 KO cultures respectively (Figure 1A), therefore it was significantly less toxic to SIRT1 KO cells than to wt MEFs in agreement with a previous study [8]. The treatment induced the expression of the UPR markers BiP and CHOP at 24 h, but we did not observe differences between the two genotypes in this response (Figure 1B).

SIRT1 is an inhibitor of mTOR [22], a potent regulator of cell metabolism and survival [24]. Therefore, we hypothesized that SIRT1-deficiency could potentiate mTOR activity and favor cell survival. We first confirmed that mTOR activity was enhanced in SIRT1 KO cells by measuring the phosphorylation of one of its substrates. As expected, 4E-BP1 was hyperphosphorylated in SIRT1−/− MEFs, shown by the accumulation of the slower migration band in the blot (γ band) (Appendix A). Tunicamycin did not affect the phosphorylation pattern of 4E-BP1 (Appendix A).

Rapamycin, the inhibitor of mTOR, caused the dephosphorylation of 4E-BP1 in SIRT1−/− (appearance of the fast migration bands β and α in Appendix A) as expected [25]. Rapamycin held cell proliferation at a low level, as reflected here by the lower number of nuclei at 48 h (Appendix A) and in agreement with previous reports [26]. Rapamycin did not induce necrosis (low LDH activity, Appendix A) or apoptosis in SIRT KO cells (Appendix A). Rapamycin pre-treatment for 24 h but not 4 h rendered the cells insensitive to tunicamycin toxicity (Appendix A). The latter induced chromatin condensation and apoptosis as expected (Appendix A). Therefore, mTOR hyperactivity was not involved in the major cell resistance to tunicamycin in SIRT1-deficient MEFs.

### 2.2. SIRT1−/− Cells Constitutively Express Heat Shock Proteins

In order to understand the mechanism underlying the lower susceptibility of SIRT1 KO cells to tunicamycin toxicity, we then studied the expression of chaperone proteins, since ER stress leads to the accumulation of unfolded proteins. Wild-type and SIRT1-deficient cells expressed similar amounts of BiP and HO-1 (Figure 1B and Figure 2A). However, Hsp70 and Hsp27 were present only in SIRT1-deficient MEFs (Figure 2A). Tunicamycin treatment reduced the expression of Hsp70 by 57% (Figure 2B). In addition, the remaining Hsp70 protein migrated faster in the polyacrylamide gel than the protein present in control conditions, suggesting that tunicamycin induced post-translational modifications besides altering Hsp70 levels. To observe this shift, the electrophoresis was run for longer in Figure 2B than in Figure 2A but the treatments were identical.

### 2.3. Hsp70 Silencing Abolished SIRT1 Resistance to Tunicamycin Toxicity

Since Hsp70 is known to protect against several types of cell insults, we hypothesized that this chaperone could be involved in the resilience of SIRT1 KO cells against ER stress. Transient transfection with plasmids expressing three different Hsp70 shRNAs did not achieve a strong reduction in Hsp70 levels at 72h post-transfection (20% to 41% decrease for individual Hsp70 shRNA and 40% for the mix of the three shRNA versus non-silencing, n = 2, data available upon request). Therefore, we generated four stable cell lines with constitutive expression of a Hsp70 shRNA or a non-silencing shRNA. Hsp70 deficiency did not significantly affect cell growth (data available upon request). Two of the clones had reduced expression of Hsp70 (155 and 156) when compared to non-silencing (NS, Figure 3A). Unexpectedly, the NS clone had higher levels of Hsp70 than the original SIRT1 KO cell line (control “C” in Figure 3A). Hsp70 silencing exacerbated tunicamycin toxicity: survival was 45% and 55% in the control clone and clone 160, respectively, while it was 70% in the NS clone and 12–16% in the clones with an efficient silencing of Hsp70 (Figure 3B top graph). Interestingly, cell resistance to tunicamycin toxicity displayed a positive correlation with Hsp70 expression levels (Figure 3B bottom graph).

### 2.4. SIRT1 Absence Did Not Impair Proteasome Activity

Since impaired clearance of proteins may stimulate the synthesis of chaperones to maintain protein folding [17,27], we analyzed proteasome activity in SIRT1 wt and KO cells in control conditions. We observed that SIRT1-deficient cells had a small but significant increase in enzymatic activity (Appendix A). Therefore, the increased expression of Hsp70 could not be attributed to a dysfunction of the proteasome. Treatment with the proteasome blocker MG132 induced the accumulation of ubiquitinated proteins in both wt and SIRT1 KO cells but the response was more pronounced in wt cells (Appendix A). As expected, MG132 induced the expression of Hsp70 in wt and SIRT1 KO cells. However, Hsp70 expression was much higher in SIRT1 KO cells (Appendix A).

### 2.5. Heat Shock Element (HSE) Was Not Involved in the Constitutive Production of Hsp70 in SIRT1−/− Cells

We next measured *Hsp70* mRNA expression in basal conditions and observed that the transcripts were 6-fold higher in SIRT1 KO cells than in wt MEFs (Figure 4A). In order to determine whether HSE participated in the transcription of Hsp70, we performed a luciferase promoter-reporter assay and observed that, in basal conditions, there were no differences between wt and KO cells in terms of HSE activity (Figure 4B). In contrast, treatments known to operate through HSE such as arsenite or heat shock demonstrated that the transcriptional activation through HSE in SIRT1 KO cells was much stronger and lasted longer than in wt cells (Figure 4B). These data suggest that canonical stimulation of Hsp70 transcription through HSE was preserved in both genotypes, but this regulatory element was not involved in the larger transcription of Hsp70 in SIRT1 KO cells under non-stimulated conditions.

### 2.6. STAT3 and Its Target Gene Socs-3 Were Upregulated in SIRT1−/− Cells

A previous study described that STAT3 binds to the human HSP70 promoter but this has not been reported for the mouse Hsp70 promoter. The transcriptional activity of STAT3 is abrogated by SIRT1-mediated deacetylation [20]. The acetylation of STAT3 correlates with its phosphorylation and both are important for its transcriptional activity [28]. We therefore examined STAT1 and STAT3 in our cultures and observed that STAT3 was present at higher amounts in SIRT1 KO cells than in wt cells (Figure 5A). In addition, STAT3 was constitutively phosphorylated on the tyrosine residue Y705 in SIRT1-deficient cells but not in wt cells (Figure 5A). Tunicamycin treatment completely abolished STAT3 phosphorylation in SIRT1−/− cells (Figure 5B). Immunofluorescence with the anti-STAT3 antibody demonstrated that STAT3 localized to the nucleus of SIRT1 KO cells in condensed spots whereas it was undetectable in wt MEFs (Figure 5C). To check whether nuclear STAT3 could be functional, we analyzed the mRNA expression of its well-known target gene *Socs-3* [29] and observed a significant increase in SIRT1 KO cells when compared to the wt cells (Figure 5C graph).

We next evaluated the binding of STAT3 to the *Socs-3* and *Hsp70* promoters.

The typical consensus sequence for STAT transcription factors is TTCCnGGAA, but STAT3 can bind alternate or incomplete sequences [30]. The analysis of its promoter showed the existence of a putative binding site on the mouse *Hspa1a* promoter (see Section 4). Chromatin immunoprecipitation showed that the efficiency of STAT3 antibodies to immunoprecipitate DNA was higher than the efficiency of the control IgG antibody but no differences were observed between wt and KO cells (Figure 5D(a,c), %input). However, the binding of STAT3 to the locus that we identified in the *Hsp70* promoter was slightly higher and statistically significant in KO vs. wt MEFs when normalized to the negative locus (Figure 5D(b), fold enrichment). In the case of the *Socs-3* promoter, the binding tended to be higher in KO than in wt MEFs although the difference was not statistically significant (Figure 5D(d)). We next tried to block STAT3 activity with Stattic, a potent inhibitor of STAT proteins. Though a concentration of 5 μM Stattic dephosphorylated STAT3 after 48 h (data available upon request), it was toxic to both SIRT1 KO and wt MEFs. These results show that STAT3 is vital for cell survival and particularly in SIRT1−/− cells, where it regulates mitochondrial respiration [31].

## 3. Discussion

In this study, we demonstrated that (1) the lack of SIRT1 induced the synthesis of Hsp70 that was involved in the resilience of SIRT1−/− against tunicamycin toxicity, (2) in SIRT1−/− cells, STAT3 was constitutively phosphorylated and its target gene *Socs-3* was upregulated, (3) lack of SIRT1 facilitated the clearance of unfolded proteins and (4) the hyperactivity of mTOR in SIRT1−/− cells did not participate in the resistance to tunicamycin toxicity.

Acetylation/deacetylation of proteins are modifications that regulate gene expression in physiological and pathological conditions. As well as histones, transcription factors also undergo these alterations that together with other post-translational modifications such as phosphorylation and SUMOylation allow the cell to adapt to an unfavorable environment [32,33]. The balance between acetylation and deacetylation of proteins may be critical enough to define a crossline between survival and death. For this reason, understanding the functioning of histone deacetylases that govern this enzymatic reaction is a necessity. All members of the sirtuins family are primarily beneficial in adverse conditions, such as ER stress. SIRT1 is not the only sirtuin involved in the cellular response to ER stress. Overexpression of SIRT2 and upregulation of SIRT3 and SIRT7 protected hepatocytes from ER stress-induced damage [34,35,36]. In contrast, the silencing of SIRT4 in immortalized human astrocytes resulted in the activation of calreticulin and the reduction in ER stress [37]. Therefore, SIRT4 could potentially be a target for pharmacological intervention with inhibitors to enhance protection against ER stress. In addition, it would be of particular interest to selectively silence SIRT proteins in astrocytes in models of CNS pathologies since inactivation of SIRT1 and SIRT2 attenuated the pro-inflammatory phenotype of astrocytes in CNS autoimmunity and spinal cord injury, respectively [5,38]. Inhibitors of class I and class II HDAC, such as valproic acid or trichostatin A (TSA), were shown to be protective in ischemic brain and neurodegenerative disorders, through the induction of BDNF, Bcl-2 and chaperones [10,11,13,39]. Heat shock proteins are cytosolic or ER resident chaperones expressed at low amounts in many cell types, and they are usually induced as part of a stress response [40]. In addition to natural stimuli, such as viral infection or heat shock that boosts the production of heat shock proteins, drugs can also be used experimentally to induce them. As indicated, HDAC inhibitors fulfill this function. Considering the potential therapeutic application of HDAC activators and inhibitors [41], we need to investigate more deeply into their mechanism of action. The relationship between class III HDAC, namely the sirtuins, and chaperones has been hardly studied. Here, we describe that SIRT1 deficiency leads to the induction of the chaperones Hsp70 and Hsp27, which are potentially able to reduce the toxicity of an acute ER stress. A recent study described that B cells isolated from tamoxifen-induced SIRT1 deficiency in mice expressed high levels of *Hspa1a* and *Hspa1b* transcripts, in agreement with our findings [42]. In contrast, another study [43] showed that SIRT1+/+ MEFs constitutively expressed Hsp70 and SIRT1−/− MEFs displayed a reduced expression. With some exceptions, primary and immortalized cell lines do not express Hsp70 in basal conditions and the chaperone is induced via cellular stress. Tomita et al. did not focus their study on the transcription factors responsible for Hsp70 expression in wt and SIRT1 KO MEFs, as we and others [8] did. Therefore, it is difficult to solve the discrepancy. We believe that wt MEFs may somehow be mildly stressed, and a heat shock factor (most likely, HSF1) may be constitutively active and could upregulate Hsp70. The deficiency in SIRT1 would impede HSF1-dependent expression of Hsp70. Interestingly, Tomita et al. showed that heat shock in SIRT1-deficient MEFs was still able to induce Hsp70 expression, suggesting that mechanisms independent of HSF1 were recruited for the transcription of the *Hsp70* mRNA. This result was in accordance with some of our data. Indeed, in the promoter-reporter assay, we also showed that a transcription factor bound to HSE after heat shock in SIRT1−/− MEFs (Figure 4B) but the magnitude of the induction of Hsp70 protein was higher in wt than in SIRT1−/− MEFs (Figure 4C).

We focused our study on Hsp70 since it was more abundant than Hsp27. In particular, we demonstrated that Hsp70 silencing exacerbated tunicamycin toxicity in SIRT1−/− MEFs, pointing out the key role of this chaperone against ER stress. Since *Hsp70* transcripts were several-fold higher in SIRT1 KO than in wt MEFs, we suspected that cis-acting transcriptional elements were involved. The *Hsp70* promoter displayed consensus sequences for HSF-1, STAT3 and Sp1 among others. Valproate was shown to deacetylate Sp1 [10]. Our data suggested that the transcription factors involved in heat shock proteins expression in SIRT1-deficient cells were different from the ones recruited during the blockade of class I and IIa HDAC. So far, Sp1 has not been reported as a substrate of SIRT1. SIRT1 has been shown to induce the expression of Hsp70 through HSF-1 activation upon heat shock [21]. We showed that in basal conditions, the activity of the HSE was similar in SIRT1+/+ and SIRT1−/− MEFs. In contrast, we found that SIRT1−/− cells showed constitutive accumulation of phosphorylated STAT3 in agreement with previous studies [28] suggesting the possibility that STAT3 was involved in Hsp70 expression in the absence of SIRT1. It has been reported that SIRT1 deacetylates and inhibits STAT3. Acetylation and phosphorylation of STAT3 work in concert to allow the binding of this transcription factor to DNA [20]. Here, we presented data suggesting that STAT3 activated the *Hsp70* promoter: (i) STAT3 was constitutively phosphorylated and present in the nucleus of SIRT1−/− cells, (ii) tunicamycin induced the dephosphorylation of P-STAT3 that matches with a drop of Hsp70 levels and (iii) ChIP-qPCR showed that STAT3 binding to the *Hsp70* promoter was slightly but significantly higher in SIRT1−/− than in SIRT1+/+ cells. Unfortunately, the pharmacological blockade of STAT3 was toxic so we could not definitively resolve whether STAT3 was the transcription factor responsible for Hsp70 expression. Interestingly, in tumor cells, a protein-protein interaction between chaperones and STAT proteins has been described and enhanced Hsp70 activity increased STAT3 activity [44,45]. Therefore, it is likely that the down-regulation or loss of function of STAT3 affect the function of Hsp70 and vice versa. Due to the complex cross-regulation between STAT3 and other proteins, the identification of the transcription factors indirectly related to SIRT1 and binding loci driving Hsp70 expression in our model require additional experiments that were beyond the scope of the present study.

We also explored the ubiquitin-proteasome system (UPS) in order to examine whether SIRT1 was involved in the clearance of misfolded proteins. The fact that the accumulation of poly-ubiquitinated proteins was lesser after the blockade of the proteasome activity in SIRT1−/− than in SIRT1+/+ cells suggested that SIRT1 delays protein degradation through the UPS. This is in agreement with the literature describing that SIRT1 inhibitors accelerate the clearance of protein aggregates and are therefore of therapeutic interest for some neurodegenerative disorders [46].

Since SIRT1 controls the activity of mTORC1 and there is a crosstalk between UPR and mTORC1 signaling pathways [23], we also analyzed mTORC1 contribution in our model. Our data showing that the mTORC1 inhibitor rapamycin abolished tunicamycin toxicity were in agreement with previous reports describing that mTORC1 stimulation potentiates ER stress-induced apoptosis [23] and discounted the hyperactivation of mTORC1 as a protective mechanism related to the lack of SIRT1.

In summary, SIRT1-deficient cells undergo transcriptional adaptation associated with the constitutive expression of heat shock proteins and are equipped to fight against proteotoxicity.

## 4. Materials and Methods

All products for cell culture unless otherwise stated were purchased from Life Technologies (Alcobendas, Spain). GeneCellin was from BioCellChallenge, (Toulon, France). Tunicamycin, rapamycin, sodium arsenite and Stattic were purchased from SIGMA, (Darmstadt, Germany). Anti-SIRT1 (07-131, 1:2000) was purchased from Millipore (Darmstadt, Germany). Anti-Hsp70 antibody (HSP01, 1:2000) was from Calbiochem (Darmstadt, Germany), anti-Hsp27 (SPA-801, 1:1000) from Enzo Life Sciences (Farmingale, New York, USA). Anti-Actin (A5228, 1:10,000) and anti-tubulin (T4026, 1:50,000) antibodies were from SIGMA and anti-Ubiquitin (ab137031, 1:2000) from Abcam (Cambridge, UK). Anti-HO-1 (OSA111, 1:1000) and anti-BiP (SPA-826, 1:2000) were from Stressgen (Victoria, BC, USA). Anti-STAT3 (610190, 1:1000) and anti-STAT1 (610186, 1:1000) were from BD Transduction Lab (Franklin Lakes, NJ, USA), anti-P-STAT3^Y705^ (9131, 1:1000), anti-P-STAT1^Y701^ (5806, 1:1000) and anti-4E-BP1 (9452, 1:2000) were from Cell Signaling Technologies (Danvers, MA, USA). Anti-CHOP (sc-7351, 1:500) was from Santa Cruz (Dallas, CA, USA). IRDye 800CW conjugated anti-mouse and anti-rabbit IgG (926-32210 and 926-32211) and IRDye 680 anti-mouse and anti-rabbit (926-32220 and 926-32221) antibodies were from LI-COR Biosciences (Lincoln, NE, USA) and diluted 1:8000. Primers were purchased from IDTConda (Torrejón de Ardoz, Madrid, Spain).

### 4.1. Cell Cultures and Treatments

Mouse embryonic fibroblasts (MEFs) SIRT1+/+ or SIRT1−/− [47] were cultured in DMEM (with 4.5 g/L glucose and glutamine) supplemented with 10% fetal bovine serum (FBS) and 1% penicillin/streptomycin for three days. The day before the experiment, cells were plated in multi-well dishes at 20,000 cells/cm^2^ unless otherwise stated.

Cells were treated with 3 μM tunicamycin for 24 h. Rapamycin was added at 0.01–2 μM for 4 h or 24 h. Stattic was added at 0.1–5 μM for 48 h. Control sister cultures were treated with vehicle.

### 4.2. Stable Cell Lines Expressing shRNA-Hsp70

SIRT1−/− MEFs were plated at 1.2 × 10^6^ cells in a 10 cm dish. Twenty-four hours after passage, cells at 50% confluency were transfected using 1.5 μg of DNA with GeneCellin as the transfection agent (1:4 ratio). Three short hairpin RNA constructs (sequences in Appendix A) against *Hsp70* (155, 156 and 160 clones) or non-silencing RNA (NS), flanked by sequences of the human micro-RNA-30 (miRNA) primary transcripts, inserted in GIPZ plasmids were used in this study (GIPZ lentiviral shRNA Hsp70 mouse, RMM4532-NM_010478, clones V2LMM_96156, V2LMM_96160, V2LMM_96155, and RHS4346 from Openbiosystems-Horizon. Huntsville, AL, USA). Polyclonal stable cell lines were generated by selection with 4.5 μg/mL puromycin and maintained in medium containing penicillin/streptomycin and fungizone to avoid contaminations during the first passages. Once the stocks were established, cells were cultured in medium containing FBS and penicillin/streptomycin only.

### 4.3. Viability Assays

Lactate dehydrogenase activity and DAPI staining were performed to assess cell death and survival respectively. Culture medium was collected and immediately processed to monitor LDH activity as previously described [25].

Cells were fixed with 4% paraformaldehyde, permeabilized with 0.5% Triton X-100 for 10 min and then incubated with 300 nM DAPI for 10 min. Images were taken under a fluorescent microscope Olympus IX71 and the number of total stained nuclei (with condensed DNA and with normal conformation DNA) were counted in 3 fields per well with the Cell^B software version 3.4 (Olympus, Hamburg, Germany). Two to three wells per condition were counted in each culture.

### 4.4. Immunofluorescence

Immunofluorescence was performed according to a standard protocol in use in our laboratory [48]. Cultures were fixed as for DAPI staining and permeabilized with 0.5% Triton X-100 for 15 min. Cells were incubated in blocking solution (3% normal goat serum in PBS) for 30 min at RT and then overnight at 4 °C with mouse anti-STAT3 antibody (Zymed 13-7000, 1:100) in 1% normal goat serum. After 1 h incubation at RT with a goat anti-mouse Alexa Fluor 488, cells were washed, maintained in PBS and images were taken as indicated in the previous section.

### 4.5. Western Blot

Cells were collected in RIPA buffer plus a Complete protease inhibitor cocktail and processed for Western blot as previously indicated [25]. HRP- or fluorophore-coupled secondary antibodies were used (SIGMA and LI-COR respectively). Blots were developed using ECL or scanned in an Odyssey infrared imaging system (LI-COR, Bad Homburg, Germany). Band intensity was measured using QuantityOne (BioRad, Barcelona, Spain) or Image Studio v5.2 (LI-COR). The ratio of the abundance of the protein of interest to actin or to tubulin was calculated and results were typically expressed as fold change versus wt at time 0 or versus control.

### 4.6. Real-Time PCR

RNA was extracted from MEFs with PureLink RNA mini kit from Ambion. One μg RNA was used for reverse transcription with a high-capacity cDNA reverse transcription kit from Applied Biosystems. cDNA was amplified via PCR with specific primers for mouse *Hsp70* (F: 5′-GGCTGATCGGCCGCAAGTT-3′; R: 5′-GGAAGGGCCAGTGCTTCAT-3′); *L14* (F: 5′-GGCTTTAGTGGATGGACCCT-3′; R: 5′-ATTGATATCCGCCTTCTCCC-3′) or *Socs*-3 (F: 5′-GCGAGAAGATTCCGCTGGTA-3′; R: 5′-CGTTGACAGTCTTCCGACAAAG-3′) (Appendix A). Amplification was carried out with SYBR green PCR master mix (Life Technologies) in a BIO-RAD C1000 thermal cycler. Optimized thermal cycling conditions were as follows: 2 min at 50 °C, 10 min at 95 °C followed by 40 cycles of 15 sec at 95 °C and 1 min at 60 °C and finally 1 min at 95 °C, 1 min and 10 s at 55 °C. Melt curves were performed upon completion of the cycles to ensure absence of non-specific products. Gene expression was calculated using the CT comparative method 2^−ΔΔCt^ normalizing to *L14* values. Results were expressed in fold change versus wt in control conditions.

### 4.7. Proteasome Activity Assay

Chymotrypsin-like and peptidyl-glutamyl-peptidase-like enzymatic activities were determined using the fluorescent peptides LLVY-AFC and Z-LL-AMC, respectively, according to a published protocol [49]. Cells were collected in lysis buffer (1 mM EDTA, 1 mM ATP, 1 mM DTT, 20% glycerol, 10 mM Tris-HCl, pH7.5), sonicated and spun at 13,500× *g* for 10 min at 4 °C. Twenty μg of proteins from the supernatant were incubated in a microtiter plate in a reaction buffer containing 0.5 mM EDTA, 50 mM Tris-HCl pH 8 and 40 μM of LLVY-AFC or Z-LL-AMC. Fluorescence was read every 2 min for 30 min at 37 °C in a Spectramax Gemini fluorimeter plate reader from Molecular Devices (Wokingham, UK; excitation/emission 400/505 nm or 380/460 nm).

Activity was calculated as Δfluorescence/min/mg protein.

Western blot for ubiquitinated proteins was performed (anti-ubiquitin, Abcam) at 6 or 15 h after 10 μM MG132 treatment. The band intensity of proteins of high molecular weight (250–75 kDa approx) was quantified and normalized to tubulin.

### 4.8. HSE-Luciferase Reporter Assay

Cells plated at 20,000 cells/well in 24-well plates, were transfected 24 h later with Pathway Profiling Plasmids (0.5 μg) containing the consensus sequences of the heat shock element (HSE) or TAL (control) promoters according to the instructions from the manufacturer (Clontech Laboratories, Mountain View, CA, USA). Forty-eight hours later, cells were collected in 100 μL of lysis buffer, span at 16,000 g at 4 °C for 30 s and an aliquot was used to quantitate protein. The activity of the promoters was determined using a dual luciferase reporter assay system (Promega, Madison, WI, USA): 20 μL of the supernatant was incubated with the reagents for 30 min and light emission was monitored in an Orion microplate luminometer (Berthold Detection Systems, Pforzheim, Germany). In pilot experiments, we noticed that different treatments altered the Renilla activity, therefore the activity of the HSE promoter was compared with that of the TAL control promoter. Promoter activity was determined in basal conditions or under conditions known to activate HSE. Cells were incubated with 20 µM sodium arsenite for 6 h. To determine the effect of delayed arsenite treatment, cultures were incubated with arsenite for 6 h and then allowed to recover in control medium for 18 h. For heat shock treatment, cells were incubated at 42 °C for 2 h, allowed to recover at 37 °C for 4 h and processed for the luciferase assay.

### 4.9. Chromatin Immunoprecipitation (ChIP)

Cells were plated at 4 × 10^6^ cells in a 15 cm dish (one dish per condition). Three days later, cells were treated with 1% methanol-free formaldehyde for 15 min at room temperature and then for 10 min with 125 mM glycine. After two washes with cold PBS, cells were collected in 10 mL PBS plus Complete, span at 1000× *g* for 5 min at 4 °C. The cell pellet was resuspended in 1 mL of nuclei buffer (Tris-HCl 50 mM pH 8.1; EDTA 10 mM and SDS 10%) and chromatin was sheared into 200–700 bp fragments for 25 min at 6 °C in a Covaris S2 ultrasonicator at intensity 2 (Brighton, UK). Sonication was performed at the Centre for Genomic Regulation (Barcelona, Spain). The sonicated extract was span 10 min at 12,000 rpm and the supernatant was collected. For each ChIP, 25 μg of DNA was used. The sample was diluted 10-fold in immunobuffer (Tris HCl 16.7 mM pH 8.1; EDTA 1.2 mM, SDS 0.01%, NaCl 165 mM, Triton X-100 1%, Complete) and pre-cleared for 1 h at 4 °C with Dynabeads^TM^ Protein G (Invitrogen Carlsbad, CA, USA). A sample corresponding to 1% of the total chromatin was collected as input DNA and stored at −80 °C. Sheared chromatin was incubated with 2 µg of anti-STAT3 (sc-483X Santa Cruz) and 1 µg anti-P-STAT3 (sc-7993X, Santa Cruz) or 3 µg of IgG normal rabbit serum (sc-2027X, Santa Cruz) for 6 h at 4 °C. In parallel, Dynabeads were blocked for 6 h at 4 °C with PBS containing 10 mg/mL of tRNA and 5 mg/mL BSA. Then, the pre-cleared chromatin/antibodies complexes were incubated with Dynabeads overnight at 4 °C. Dynabeads were then washed twice for 4 min at 4 °C successively with the following buffers: low salt (HEPES 50 mM pH 8.1; EDTA 1 mM, Deoxycholate 0.1%, NaCl 140 mM, Triton X-100 1%), high salt (HEPES 50 mM pH 8.1; EDTA 1 mM, Deoxycholate 0.1%, NaCl 500 mM, Triton X-100 1%), Li buffer (Tris HCl 10 mM pH 8.1; EDTA 1 mM, deoxycholate 0.5%, LiCl 250 mM, NP-40 0.5%) and finally washed once with TE buffer (Tris HCl 10 mM pH 8.1; EDTA 1 mM). DNA was then reverse crossed-linked, eluted and extracted with the iPURE kit from Diagenode (Liège, Belgium) according to the instructions from the manufacturer.

In order to identify a putative STAT3 binding site in the promoter of *Hsp70*, we searched a sequence similar to the one described by Yamagishi [19]: between −206 and −187 in the human HSP70 promoter (CTGGAATATTCCCG) and found the sequence CTGGAAGATTCCT between −107 and −95 in the mouse hspa1a promoter. Underlined sequences correspond to STAT3 consensus sequences in the human and mouse *Hsp70* promoters. 

Immunoprecipitated DNA was assayed for real-time PCR as described above with the following primers (Appendix A): positive locus STAT3 on *Hsp70* Pr: −143/+7 [15] (F: 5′-AGGGAGGCGGGGAAGCTCC-3′ and R: 5′-GTCTGGTGACCTGCTCGCCG-3′, amplicon 150 bp), positive locus STAT3 on *Socs3* Pr −1247/−1039 (F: 5′-CCCCCAACTTCTCATTCACA-3′ and R: 5′-TACATGAGGACCTCGGAGTG-3′, amplicon 208 bp) [29] and negative locus STAT3 on *Hsp70* Pr: −1164/−1001 (F: 5′-AGAAGAAATGGGGCTGGACG-3′ and R: 5′-TCCGGAGTGCTGGAATCCTA-3′, amplicon 163 bp). Data were represented as percentage of input (%input) and fold enrichment.

For %input, results were expressed as percentage of input for the DNA immunoprecipitated (Ct DNA) either with IgG or with the antibodies of interest (P-STAT3+STAT3) and calculated as follows:%input = 100 × 2^[(CtInput − Log2 (Input dilution factor)) − Ct DNA]^

Whereas, fold enrichment was calculated as follows:2^(−ΔΔCt)^ in which ΔΔCt = ΔCt_positive_ − ΔCt_negative_.

ΔCt_positive_ = [Ct DNA − (CtInput − Log2 (input dilution factor))] was obtained with positive locus STAT3 primers and ΔCt_negative_ = [Ct DNA − (CtInput − Log2 (input dilution factor))] was obtained with negative locus STAT3 primers. In that case, Ct DNA refers to the DNA immunoprecipitated with the P-STAT3+STAT3 antibodies.

### 4.10. Statistical Analysis

Samples (n) were individual cultures used to perform the experiments. Data are represented as mean ± SEM. To compare two groups, two-tailed Student’s *t*-test was used. For the comparison of three or more groups, one-way ANOVA followed by the Bonferroni multiple comparison test was used. Equal variance was checked with Bartlett’s test. When data did not follow a normal distribution or variance was different between groups, the Kruskall-Wallis or Mann-Whitney non-parametric tests were applied. To analyze the effect of two independent variables, two-way ANOVA followed by the Šídák multiple comparison test was used. All data were analyzed using GraphPad Prism 9.

## Figures and Tables

**Figure 1 ijms-25-02856-f001:**
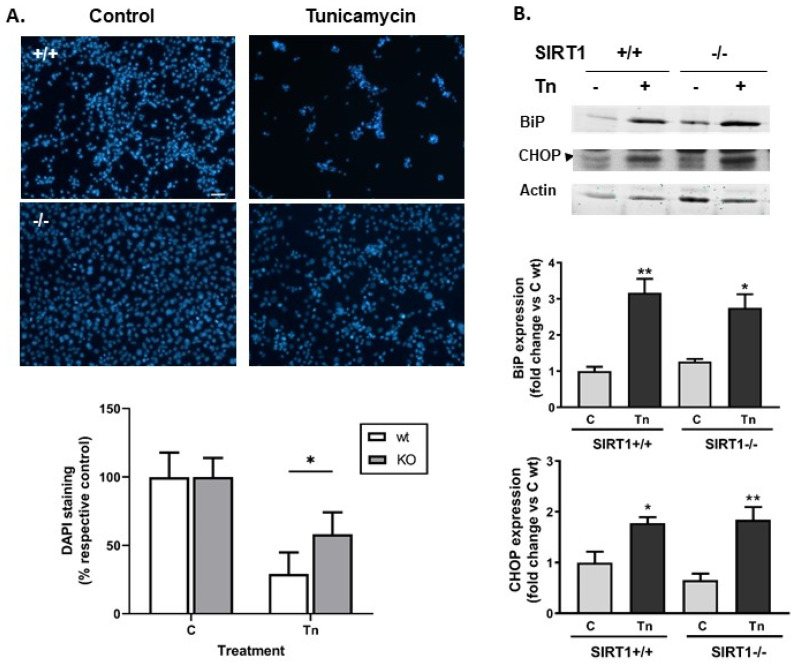
SIRT1−/− MEFs were more resistant to tunicamycin-induced death than SIRT1+/+ MEFs. (**A**). Representative microphotographs of cells stained with DAPI 24 h after the treatment with 3 μM tunicamycin. Scale bar 100 μm. Graph: quantitative representation of surviving cells after the lesion. Nuclei stained with DAPI were counted and expressed as a % of respective control. Results are mean ± SEM, n = 5–7 cultures, * *p* < 0.05. (**B**). Western blot of BiP and CHOP after tunicamycin treatment in wt and SIRT1 KO cells. Arrowhead indicates the band corresponding to CHOP. Other bands are not specific. Semi-quantitative analysis of BiP and CHOP was determined by calculating the ratio of the protein of interest to actin. Results from 3 different experiments are expressed as mean ± SEM. * *p* < 0.05, ** *p* < 0.01 indicate statistical significance vs. respective control.

**Figure 2 ijms-25-02856-f002:**
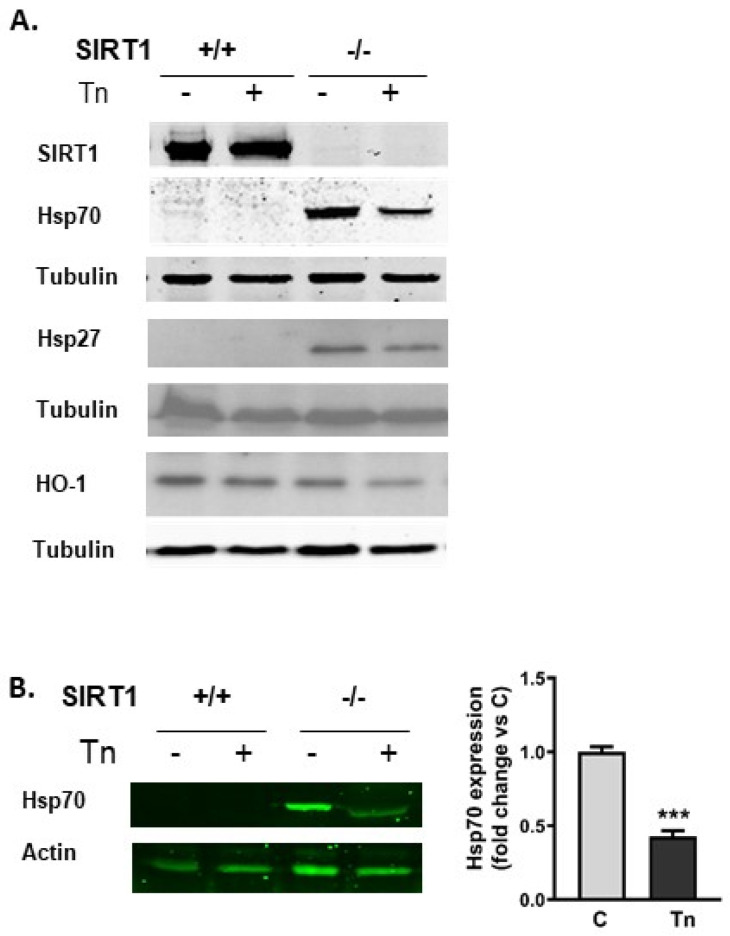
SIRT1 deficiency induced the expression of heat shock proteins. (**A**). Western blot of SIRT1, Hsp70, Hsp27 and HO-1 in wt and SIRT1 KO cells show that Hsp70 and Hsp27 were not expressed in SIRT1+/+ but constitutively synthesized in SIRT1−/− cells. HO-1 was similarly expressed in wt and SIRT1 KO cells. Tunicamycin reduced Hsp70 expression in SIRT1−/− cells. (**B**). Western blot and quantitative analysis of Hsp70 after tunicamycin treatment in SIRT1−/− cells. Results were measured as the ratio of Hsp70 to actin and expressed as mean ± SEM (n = 3) *** *p* < 0.001.

**Figure 3 ijms-25-02856-f003:**
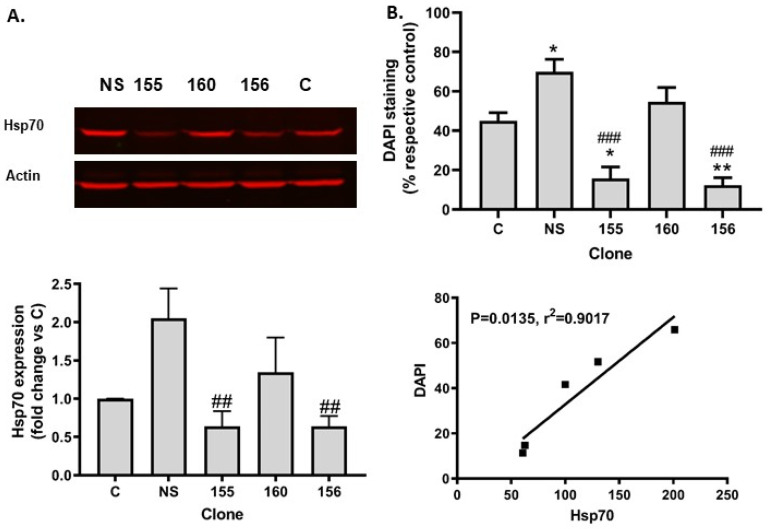
Hsp70 silencing in SIRT1-deficient cells abolished resistance to tunicamycin toxicity (**A**). Western blot of Hsp70 in the control SIRT1 KO cell line and in the stable polyclonal cell lines expressing non-silencing RNA (NS) or 3 different Hsp70 shRNA (155, 156 and 160). Top: representative blot of Hsp70 expression. Bottom graph: quantitative representation of Hsp70 expression. Results are mean ± SEM, n = 4–5. Note that only shRNA 155 and 156 efficiently repressed Hsp70 synthesis. (**B**). Quantitative analysis of surviving cells 24 h after tunicamycin treatment. Top graph: DAPI+ cells were counted and expressed as % of respective control cells. Results are mean ± SEM, n = 4–5. Bottom graph: linear regression analysis shows that protein levels of Hsp70 correlated with cell resistance to tunicamycin toxicity. * *p* < 0.05, ** *p* < 0.01 vs. control; ## *p* < 0.01, ### *p* < 0.001 vs. NS.

**Figure 4 ijms-25-02856-f004:**
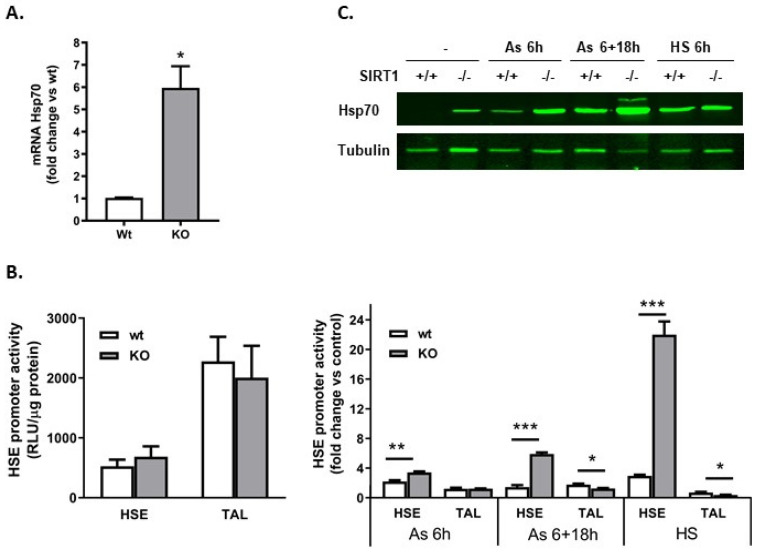
Hsp70 was upregulated in SIRT1 KO cells. Transcriptional activation of HSE was not involved in the effect. (**A**) Quantitative RT-PCR of *Hsp70*. SIRT1+/+ and SIRT1−/− cells were scrapped in basal conditions and processed for mRNA determination using SYBRGreen. *Hsp70* mRNA levels were normalized with *L14*. Results calculated as fold change vs. wt are presented as mean ± SEM. n = 4, * *p* < 0.05. (**B**) HSE basal activity was similar in wt and KO cells, but upon stress conditions, the HSE promoter activity was stronger and lasted longer in KO than in wt cells. MEFs were transfected with pTAL or pHSE plasmids and 48 h later, cells were lyzed and luciferase activity was monitored. Results, calculated as RLU/µg protein, are expressed as mean ± SEM (left graph, n = 3). Sister cultures were either incubated at 42 °C (HS) or with arsenite (As) for 6 h or 6 h + 18 h recovery and processed for luciferase activity (right graph). Results, calculated as fold increase vs. respective control, are expressed as mean ± SEM (n = 3). * *p* < 0.05, ** *p* < 0.01 and *** *p* < 0.001. (**C**) Western blot of Hsp70 after chemical stress (As) or heat shock (HS) treatments performed as in (**B**).

**Figure 5 ijms-25-02856-f005:**
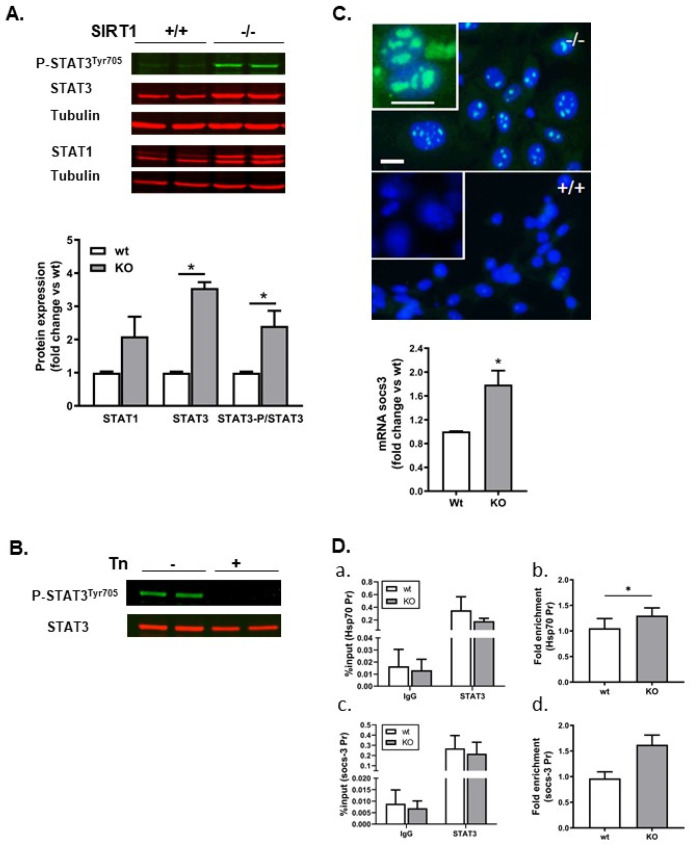
STAT3 was constitutively phosphorylated and accumulated in the nucleus of SIRT1−/− cells. It was recruited to *Hsp70* and *Socs-3* promoters. (**A**) Western blot of STATs proteins in basal conditions showing that STAT3 was upregulated in SIRT1−/− cells. In addition, P-STAT3^Tyr705^ was present only in SIRT1 KO cells. Graph: quantitative expression of STAT1, STAT3 and P-STAT3. Results are mean ± SEM, n = 4. * *p* < 0.05. (**B**) Western blot of P-STAT3^Tyr705^ after tunicamycin treatment (Tn). Cells were collected 24h after Tn addition and processed for western blot. (**C**) Immunofluorescence showed STAT3 accumulation in the nucleus of SIRT1−/− but not in SIRT1+/+ MEFs in basal conditions. Scale bar 20 µm. qRT-PCR showed that *Socs-3* expression was significantly higher in SIRT1−/− than in SIRT1+/+ cells. Results calculated as fold change vs. wt are presented as mean ± SEM, n = 4. * *p* < 0.05 vs. wt. (**D**) Chromatin immunoprecipitation analysis of *Hsp70* and *Socs-3* promoters was carried out in SIRT1+/+ and SIRT1−/− MEFs. Chromatin was immunoprecipitated with anti-P-STAT3/anti-STAT3 antibodies or rabbit IgG serum, DNA was amplified with primers that bound to the region between −143 and +4 bp (that overlapped the putative STAT3 binding site) on the *Hsp70* promoter or with primers that bound to the region between −1247 and −1039 bp on the *Socs-3* promoter. The amplification with primer pairs that bound to the region between −1164 bp and −1001 bp on the *Hsp70* promoter were used to normalize the data. Results are expressed as a percentage of input for the negative antibody (IgG) and the antibodies of interest (P-STAT3/STAT3) (**a**,**c**), and fold enrichment normalized to a region far away (−1164 to −1001) from the initiation transcription site on the *Hsp70* promoter (**b**,**d**). Results are mean ± SEM, n = 4. * *p* < 0.05.

## Data Availability

Data are contained within the article and Appendix A.

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
