# Peer review of "Lack of the Histone Deacetylase SIRT1 Leads to Protection against Endoplasmic Reticulum Stress through the Upregulation of Heat Shock Proteins"

_ijms, 2024, doi:10.3390/ijms25052856_

Round 1
Reviewer 1 Report
Comments and Suggestions for Authors
The research by Jèssica Latorre et al. investigates the SIRT1 deficiency-induced ER stress resistance in mouse MEF cells and identifies upregulation of HSP70, possibly via STAT3 over-activation. The manuscript is well-written, with an informative introduction, clear results, and a convincing discussion. However, certain experimental designs do not fully support the conclusions drawn, and key controls are absent. Additional experiments are needed for results validation and to strengthen the conclusions. The methods section requires more detail, and minor labeling issues are present. Detailed comments are as follows:
Major comments:
1. Figure 2, it’s unclear whether the increase of Hsp70 is due to the off-target effect. To exclude this possibility, overexpress SIRT1 in the SIRT1 -/- cell to see whether the Hsp70 level would be affected.
2. Figure 3A, it would be helpful to strengthen the result with qRT-PCR verification of HSP70 gene knockdown efficiency.
3. Figure 3B, it would be helpful to strengthen the result by adding the non-treated experiment groups to show the reduction of DAPI staining is not due to HSP70 knockdown.
4. Figure 4B, other than only measuring the global HSE activity, validating whether HSF1 binds to the HSEs of the HSP70 promoter by ChIP-qPCR would be more accurate.
5. Figure 5C, the STAT3 immunofluorescence signal looks abnormal, it would be helpful to validate the signal such as by immunofluorescence staining the p-STAT3.
6. Figure 5C, why was the nuclear of SIRT1-/- much bigger than SIRT1 +/+?
7. Figure 5D, the KO group has a huge amount of STAT3 signal in the nuclear in Figure 5C, however, the STAT3 ChIP-qPCR signal was almost comparable to the wt. Does this mean the STAT3 signal was not real in Figure 5C? or does this suggest the nuclear STAT3 has no function?
8. Please explain what does “immunoprecipitated with anti-P-STAT3/anti-STAT3 antibodies” mean? And please explain what’s the reason for doing the ChIP in this way.
9. To confirm the legitimacy of the ChIP-qPCR signal for STAT3, include analyses of both positive and negative binding regions, along with IgG ChIP-qPCR controls for these regions, in the same figure.
Minor comments:
1. Figure 2A, SIRT1 western blot didn’t match with the original image. Please verify.
2. Takuya Tomita et al. 2015 showed that SIRT1-/- MEF cells have reduced Hsp70. However, in this research, SIRT1-/- MEF cells have increased Hsp70. It would be valuable to discuss potential reasons for this discrepancy. Addressing this point will strengthen the interpretation of the results.
3. Lines 351-359, detailed information regarding the “Stable cell lines expressing shRNA-HSP70”, such as shRNA sequence, expression system et al. is missing. The author didn’t maintain the cell line with puromycin, the potential issue is loss of expression of shRNA during passaging.
4. The two panels for Figure 4B should use the same Y-axis.
5. Figure 5C, scale bar missing.
Author Response
Dear reviewer,
Please see the attachment.
Best regards,
Valérie Petegnief

Reviewer 2 Report
Comments and Suggestions for Authors
In this study, the researchers primarily investigated the changes in heat shock protein levels in the context of histone deacetylase SIRT1 deficiency. The findings revealed that SIRT1 deficiency leads to the upregulation of chaperone proteins and enhanced proteasome activity. This response is beneficial for cells in coping with endoplasmic reticulum stress. Their findings offer new insights and, to some extent, provide potential avenues for the treatment of related diseases. These results hold certain reference value for future research and clinical applications.
This study is interesting. But I have some recommendations and questions.
Recommendations and questions:
1. It would be better if a summary diagram of the key pathways is attached to the manuscript.
2. There are slight formatting issues with this article, such as incorrect use of spaces in certain areas and inconsistent font size in ‘Methods’.
3. Several abbreviations must be spelled out for the first time in the introduction.
4. The research focus of this article is on the SIRT1 protein. However, it is important to note that the SIRT family consists of various protein types, and their roles and mechanisms of action may differ. While this study specifically investigates the changes associated with SIRT1 deficiency, it does not directly address the involvement of other proteins in the SIRT family in the proposed mechanism. Is there any other protein that has the potential to become a target for inhibiting endoplasmic reticulum stress? Further research and investigation might be required.
5. In the results section, it was observed that the levels of internal reference protein were significantly uneven. Additionally, some proteins exhibited heterobands, which could potentially affect the interpretation of the results (e.g., CHOP in Fig. 1). If there are more reliable result graphs available, it is recommended to replace the existing ones with the more reliable representations.
6. The conclusion and discussion of this article primarily revolve around the mechanisms related to SIRT1, heat shock proteins, and endoplasmic reticulum stress. It would be beneficial to provide more exploration and prospects based on the disease mechanism and drug development potential of this target protein, making it easier to implement the results of this study.
7. It is suggested that the use of the Student's t-test for post hoc analysis in the context of one-way ANOVA may not be sufficiently appropriate. To address this concern, the results were either corrected using multiple comparisons methods, such as Bonferroni's correction, or re-analyzed with a more suitable post-hoc analysis method. If there is a statistician available within the author's group, it is recommended to consult and involve them in the analysis.
8. The format of doi and the names of the journals in some references are inconsistent.
Author Response
Dear reviewer,
Please see the attachment,
Best regards,
Valérie Petegnief

Reviewer 3 Report
Comments and Suggestions for Authors
The manuscript entitled “ Lack of the Histone Deacetylase SIRT1 leads to Protection against Endoplasmic Reticulum Stress through the Upregulation of Heat Shock Proteins” is significant in this field of interest. The manuscript is well-structured with enough data. However, this manuscript has minor issues that need to be addressed. Thus I recommend this manuscript for a minor revision.
1. Figures 1A and 1B exhibit significant differences, but the indications are confusing. Please review and address.
2. Merge graphs in Figures 1A (2 graphs) and 1B (2 graphs) to create a single graph for both, labeled as figures A, B, C, and D.
3. Ensure uniform color coding in the graph.
4. Highlight the novelty and significance of the study in the introduction section.
5. Incorporate Figures S1 and S2 into the main manuscript text with respective legends.
6. Simplify Figure S1 by clarifying subtitles like C (1) or C (a).
7. Merge DAPI cell count figures in S1C (first) for both 4h and 24h into one graph. Transform the remaining two figures in S1 into bar charts for easy comparison between control Rap 4 and tunicamycin Rap 4. Reorganize graphs.
8. Ensure concise subtitles in the result section, avoiding full lines.
9. Conduct a thorough typographical error check on the manuscript.
10. Shorten the lengthy discussion section paragraph (lines 254-307).
11. Improve the discussion section by comparing present findings with previous studies.
12. Include references for the protocols of Immunofluorescence and Proteasome activity assay.
13. Create a table in the supplementary section listing all primers used in the study, including those for Immunoprecipitated analysis, with proper references.
14. Provide RT-PCR operating conditions in the methods section.
15. If possible, present the quantification of Figure 2A as a new graph.
16. Include references for the protocols of Immunofluorescence and Proteasome activity assay.
Author Response

(The authors gave the same response as above.)

Round 2
Reviewer 1 Report
Comments and Suggestions for Authors
I believe the revisions have significantly improved the manuscript, and it now presents its findings more clearly and effectively. I have no more comments.
Reviewer 2 Report
Comments and Suggestions for Authors
I have no more comments.